# Dissecting graph measures performance for node clustering in LFR parameter space

## Abstract

Graph measures can be used for graph node clustering using metric clustering algorithms. There are multiple measures applicable to this task, and which one performs better is an open question. We study the performance of 25 graph measures on generated graphs with different parameters. While usually measure comparisons are limited to general measure ranking on a particular dataset, we aim to explore the performance of various measures depending on graph features. Using an LFR graph generator, we create a dataset of ∼7500 graphs covering the whole LFR parameter space. For each graph, we assess the quality of clustering with $k$-means algorithm for every considered measure. We determine the best measure for every area of the parameter space. We find that the parameter space consists of distinct zones where one particular measure is the best. We analyze the geometry of the resulting zones and describe it with simple criteria. Given particular graph parameters, this allows us to choose the best measure to use for clustering.

## 1 Introduction

Graph node clustering is one of the central tasks in graph structure analysis. It provides a partition of nodes into disjoint clusters, which are groups of nodes that are characterized by strong mutual connections. It can be of practical use for graphs representing real-life systems, such as social networks or industrial processes. Clustering allows to infer some information about the system: the nodes of the same cluster are highly similar, while the nodes of different clusters are dissimilar. The technique can be applied without any labeled data to extract important insights about a network.

There are different approaches to clustering, including ones based on modularity optimization (Newman & Girvan, 2004; Blondel et al., 2008), label propagation algorithm (Raghavan et al., 2007; Barber & Clark, 2009), Markov cluster process (Van Dongen, 2000; Enright et al., 2002), and spectral clustering (Von Luxburg, 2007). In this work, we use a different approach based on choosing a closeness measure on a graph, which allows one to use any metric clustering algorithm (e.g., Yen et al., 2009).

The choice of the measure significantly affects the quality of clustering. Classical measures are the *Shortest Path* (Buckley & Harary, 1990) and the *Commute Time* (Göbel & Jagers, 1974) distances. The former is the minimum number of edges in a path between a given pair of nodes. The latter is the expected number of steps from one node to the other and back in a random walk on the graph. There is a number of other measures, including recent ones (e.g., Estrada & Silver, 2017; Jacobsen & Tien, 2018), many of them are parametric. Despite the fact that graph measures are compatible with any metric algorithm, in this paper we restrict ourselves to the kernel $k$-means algorithm (e.g., Fouss et al., 2016).

We base our research on a generated set of graphs. There are various algorithms to generate graphs with community structures. The well-known ones are the Stochastic Block Model (Holland et al., 1983) and Lancichinetti–Fortunato–Radicchi benchmark (Lancichinetti et al., 2008) (hereafter, LFR). The first one is an extension of the Erdős–Rényi model with different intra- and intercluster probabilities of edge creation. The second one involves power law distributions of node degrees and community sizes. There are other generation models, e.g., Naive Scale-free Clustering (Pasta & Zaidi, 2017). We choose the LFR model: although it misses some key properties of real graphs, like diameter or the clustering coefficient, this model has been proven to be effective in meta-learning (Prokhorenkova, 2019).

There are a lot of measure benchmarking studies considering node classification and clustering for both generated graphs and real-world datasets (Fouss et al., 2012; Sommer et al., 2016; 2017; Avrachenkov et al., 2017; Ivashkin & Chebotarev, 2016; Guex et al., 2018; 2019; Aynulin, 2019a;b; Courtain et al., 2020; Leleux et al., 2020), etc. Despite a large number of experimental results, theoretical results are still a matter of the future. One of the most interesting theoretical results on graph measures is the work by Luxburg et al. (2010), where some unattractive features of the Commute Time distance on large graphs were explained theoretically, and a reasonable amendment was proposed to fix the problem. Beyond the complexity of such proofs, there is still very little empirical understanding of what effects need to be proven. Our empirical work has two main differences from the previous ones. First, we consider a large number of graph measures, which for the first time gives a fairly complete picture. Second, unlike these studies concluding with a global leaderboard, we are looking for the leading measures for each set of the LFR parameters.

We aim to explore the performance of of the 25 most popular measures in the graph clustering problem on a set of generated graphs with various parameters. We assess the quality of clustering with every considered measure and determine the best measure for every region of the graph parameter space.

Our contributions are as follows:

- We generate a dataset of $\sim$7500 graphs covering all parameter space of LFR generator;
- We consider a broad set of measures and rank measures by clustering performance on this dataset;
- We find the regions of certain measure leadership in the graph parameter space;
- We determine the graph features that are responsible for measure leadership;
- We check the applicability of the results on real-world graphs.

Our framework for clustering with graph measures as well as a collected dataset are available on `link_is_not_available_during_blind_review`.

## 2 DEFINITIONS

### 2.1 KERNEL $k$-MEANS

The original $k$-means algorithm (Lloyd, 1982; MacQueen et al., 1967) clusters objects in Euclidean space. It requires coordinates of the objects to determine the distances between them and centroids. The algorithm can be generalized to use the degree of closeness between the objects without defining a particular space. This technique is called *the kernel trick*, usually it is used to bring non-linearity to linear algorithms. The algorithm that uses the kernel trick is called kernel $k$-means (see, e.g., Fouss et al., 2016). For graph node clustering scenario, we can use graph measures as kernels for the kernel $k$-means.

Initially, the number of clusters is known and we need to set initial state of centroids. The results of the clustering with $k$-means are very sensitive to it. Usually, the algorithm runs several times with different initial states (trials) and chooses the best trial. There are different approaches to the initialization; we consider three of them: random data points, $k$-means++ (Arthur & Vassilvitskii, 2006), and random partition. We combine all these strategies to reduce the impact of the initialization strategy on the result.

### 2.2 CLOSENESS MEASURES

For a given graph $G$, $V(G)$ is the set of its vertices and $A$ is its adjacency matrix. A *measure* on $G$ is a function $\kappa \colon V(G) \times V(G) \to \mathbb{R}$, which gets two nodes and returns closeness (bigger means closer) or distance (bigger means farther).

A *kernel on a graph* is a graph nodes' closeness measure that has an inner product representation. Any symmetric positive semidefinite matrix is an inner product matrix (also called Gram matrix). A kernel matrix $K$ is a square matrix that contains similarities for all pairs of nodes in a graph.

To use kernel $k$-means, we need kernels. Despite that not all closeness measures we consider are Gram matrices, we treat them as kernels. The applicability of this approach was confirmed in Fouss et al. (2016). For the list of measures bellow, we use the word "kernel" only for the measures that satisfy the strict definition of kernel.

Classical measures are *Shortest Path* distance (Buckley & Harary, 1990) (SP) and *Commute Time* distance (Göbel & Jagers, 1974) (CT). SP is the minimum number of edges in a path between a given pair of nodes. CT is the expected lengths of random walks between two nodes. SP and CT are defined as distances, so we need to transform them into similarities to use as kernels. We apply the following distance to closeness transformation (Chebotarev & Shamis, 1998a; Borg & Groenen, 2005):

$$K = -H\mathcal{D}H;\ H = I - E/n,\tag{1}$$

where $\mathcal{D}$ is a distance matrix, $E$ is the matrix of ones, $I$ is the identity matrix, and $n$ is the number of nodes.

In this paper, we examine 25 graph measures (or, more exactly, 25 parametric families of measures). We present these measures grouped by type similarly to (Avrachenkov et al., 2017):

- Adjacency Matrix $A$ based kernels and measures.
    - *Katz kernel*: $K_\alpha^{\text{Katz}} = (I - \alpha A)^{-1}$, $0 < \alpha < \rho^{-1}$, where $\rho$ is the spectral radius of $A$. (Katz, 1953) (also known as Walk proximity (Chebotarev & Shamis, 1998b) or von Neumann diffusion kernel (Kandola et al., 2003; Shawe-Taylor & Cristianini et al., 2004)).
    - *Communicability kernel* $K_t^{\text{Comm}} = \text{expm}(tA)$, $t > 0$, where $\text{expm}$ means matrix exponential (Fouss et al., 2006; Estrada & Hatano, 2007; 2008).
    - *Double Factorial closeness*: $K_t^{\text{DF}} = \sum_{k=0}^{\inf} \frac{t^k}{k!!} A^k$, $t > 0$ (Estrada & Silver, 2017).
- Laplacian Matrix $L = D - A$ based kernels and measures, where $D = \text{Diag}(A \cdot \mathbf{1})$ is the degree matrix of $G$, $\text{Diag}(\mathbf{x})$ is the diagonal matrix with vector $\mathbf{x}$ on the main diagonal.
    - *Forest kernel*: $K_t^{\text{For}} = (I + tL)^{-1}$, $t > 0$ (also known as Regularized Laplacian kernel) (Chebotarev & Shamis, 1995).
    - *Heat kernel*: $K_t^{\text{Heat}} = \text{expm}(-tL)$, $t > 0$ (Chung & Yau, 1998).
    - *Normalized Heat kernel*: $K_t^{\text{NHeat}} = \text{expm}(-t\mathcal{L})$, $\mathcal{L} = D^{-\frac{1}{2}} L D^{-\frac{1}{2}}$, $t > 0$ (Chung, 1997).
    - *Absorption kernel*: $K_t^{\text{Abs}} = (tA + L)^{-1}$, $t > 0$ (Jacobsen & Tien, 2018).
- Markov Matrix $P = D^{-1}A$ based kernels and measures.
    - *Personalized PageRank closeness*: $K_\alpha^{\text{PPR}} = (I - \alpha P)^{-1}$, $0 < \alpha < 1$ (Page et al., 1999).
    - *Modified Personalized PageRank*: $K_\alpha^{\text{MPPR}} = (I - \alpha P)^{-1} D^{-1} = (D - \alpha A)^{-1}$, $0 < \alpha < 1$ (Kirkland & Neumann, 2012).
    - *PageRank heat closeness*: $K_t^{\text{HPR}} = \text{expm}(-t(I - P))$, $t > 0$ (Chung, 2007).
    - *Randomized Shortest Path distance*. Using $P$ and the matrix of the SP distances $C$ first get $Z$ (Yen et al., 2008):

    $$W = P \circ \exp(-\beta C);\ Z = (I - W)^{-1}.\tag{2}$$

    Then $S = (Z(C \circ W)Z) \div Z$; $\bar{C} = S - \mathbf{e}\,\text{diag}(S)^T$, and finally, $\mathcal{D}_{\text{RSP}} = (\bar{C} + \bar{C}^T)/2$. Here $\circ$ and $\div$ are element-wise multiplication and division. Kernel version $K^{\text{RSP}}(t)$ can be obtained with equation 1.
    - *Free Energy distance*. Using $Z$ from equation 2: $Z^h = Z\,\text{Diag}(Z)^{-1}$; $\Phi = -1/\beta \log Z^h$; $\mathcal{D}_{\text{FE}} = (\Phi + \Phi^T)/2$ (Kivimäki et al., 2014). Kernel version $K^{\text{FE}}(t)$ can be obtained with equation 1.
- Sigmoid Commute Time kernels.
    - *Sigmoid Commute Time kernel*:

    $$K_t^{\text{SCT}} = \sigma(-tK^{\text{CT}}/\text{std}(K^{\text{CT}})),\ t > 0,\tag{3}$$

    where $\sigma$ is an element-wise sigmoid function $\sigma(x) = 1/(1 + e^{-x})$ (Yen et al., 2007).

Table 1: Short names of considered kernels and other measures.

| Family | Short name | | Full name |
| | Plain measure | Logarithmic measure | |
|---|---|---|---|
| Adjacency matrix based kernels | Katz | logKatz | Katz kernel |
| | Comm | logComm | Communicability kernel |
| | DF | logDF | Double Factorial closeness |
| Laplacian based kernels | For | logFor | Forest kernel |
| | Heat | logHeat | Heat kernel |
| | NHeat | logNHeat | Normalized Heat kernel |
| | Abs | logAbs | Absorption kernel |
| Markov matrix based kernels and measures | PPR | logPPR | Personalized PageRank closeness |
| | MPPR | logMPPR | Modified Personalized PageRank |
| | HPR | logHPR | PageRank heat closeness |
| | RSP | - | Randomized Shortest Path kernel |
| | FE | - | Free Energy kernel |
| Sigmoid Commute Time | SCT | - | Sigmoid Commute Time kernel |
| | SCCT | - | Sigmoid Corrected Commute Time kernel |
| | SP-CT | - | linear combination of SP and CT |

- *Sigmoid Corrected Commute Time kernel.* First of all, we need the Corrected Commute Time kernel (Luxburg et al., 2010):

$$K^{\text{CCT}} = HD^{-\frac{1}{2}} M(I - M)^{-1} M D^{-\frac{1}{2}} H; \ M = D^{-\frac{1}{2}} \Big( A - \frac{\vec{\mathbf{d}}\vec{\mathbf{d}}^T}{\text{vol}(G)} \Big) D^{-\frac{1}{2}}$$

where $H$ is the centering matrix $H = I - E/n$, $\vec{\mathbf{d}}$ is the vector of diagonal elements of $D$ and $\text{vol}(G)$ is the sum of all elements of $A$. Then, apply equation 3 replacing $K^{\text{CT}}$ with $K^{\text{CCT}}$ to obtain $K^{\text{SCCT}}$.

Occasionally, element-wise logarithm is applied to the resulting kernel matrix (Chebotarev, 2013; Ivashkin & Chebotarev, 2016). We apply it to almost all investigated measures and consider the resulting measures separately from their plain versions (see Table 1). For some measures, like Forest kernel, this is well-known practice (Chebotarev, 2013), while for others, like Double Factorial closeness, this transformation, to the best of our knowledge, is applied for the first time. The considered measures and their short names are summarized in Table 1.

## 3 DATASET

We collected a paired dataset of graphs and the corresponding results of clustering with each measure mentioned in Table 1. In this section, we describe the graph generator, the sampling strategy, the calculated graph features, and the pipeline for the measure score calculation.

We use Lancichinetti–Fortunato–Radicchi (LFR) graph generator. It generates non-weighted graphs with ground truth non-overlapping communities. The model has mandatory parameters: the number of nodes $n$ ($n > 0$), the power law exponent for the degree distribution $\tau_1$ ($\tau_1 > 1$), the power law exponent for the community size distribution $\tau_2$ ($\tau_2 > 1$), the fraction of intra-community edges incident to each node $\mu$ ($0 \le \mu \le 1$), and either minimum degree (min degree) or average degree (avg degree). There are also extra parameters: maximum degree (max degree), minimum community size (min community), maximum community size (max community). Not the whole LFR parameter space corresponds to common real-world graphs; most of such graphs are described with $\tau_1 \in [1, 4]$ and $\mu < 0.5$ (e.g., Fotouhi et al., 2019). However, there is also an interesting case of bipartite/multipartite-like graphs with $\mu > 0.5$. Moreover, many of the datasets studied in Section 5 have $\tau_1 > 4$. Our choice is to consider the entire parameter space to cover all theoretical and practical cases.

For the generation, we consider $10 < n < 1500$. It is impossible to generate a dataset with a uniform distribution of all LFR parameters, because $\tau_1$ and $\tau_2$ parameters are located on rays. We transform

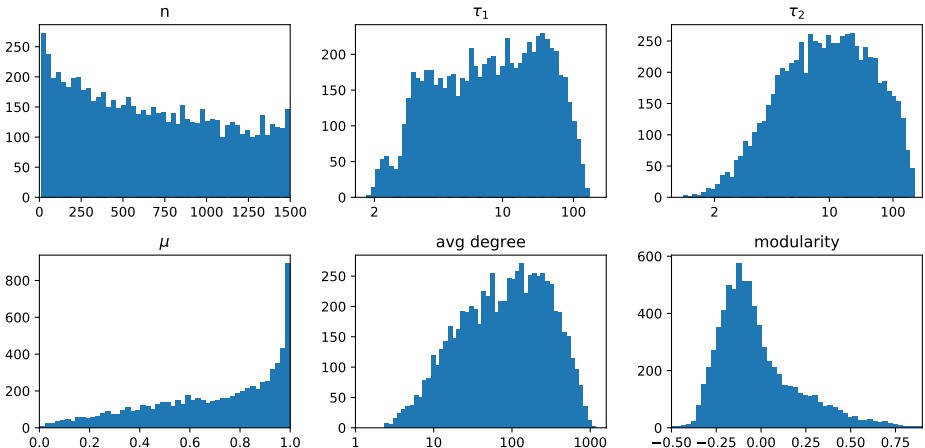

Figure 1: Distribution of graph features in the dataset

$\tau_1$ and $\tau_2$ to $\tilde{\tau}_i = 1 - (1/\sqrt{\tau_i}), i = 1, 2$ to bring their scope to the $[0, 1]$ interval. In this case, "realistic" settings with $\tau_1 \in [1, 4]$ take up 50% of the variable range. Also, as avg degree feature is limited by the $n$ of a particular graph, we decided to replace it with density (avg degree$/(n-1)$). It is not dependent on $n$ and belongs to $[0, 1]$. Using all these considerations, we collected our dataset by uniformly sampling parameters for LFR generator from the set $[n, \tilde{\tau}_1, \tilde{\tau}_2, \mu, \text{density}]$ and generating graphs with these parameters. Additionally, we filter out all disconnected graphs.

In total, we generated 7396 graphs. It is worth noting that the generator fails for some sets of parameters, so the resulting dataset is not uniform (see Fig. 1). In our study, non-uniformity is not a very important issue, because we are interested in local effects, not global leadership. Moreover, true uniformity for LFR parameter space is impossible, due to the unlimited scope of parameters.

For our research, we choose a minimum set of the features that describe particular properties of graphs and are not interchangeable.

The LFR parameters can be divided in three groups by the graph properties they reflect:

- The size of the graph and the communities: $n$, $\tau_1$, min community, max community;

- The density and uniformity of the node degrees distribution: $\tau_2$, min degree, avg degree, max degree. As avg degree depends on $n$, it is distributed exponentially, so we use $\log(\text{avg degree})$ instead;

- The cluster separability: $\mu$. As $\mu$ parameter considers only the ratio between the number of inter-cluster edges and the number of nodes but ignores overall density, we use modularity (Newman & Girvan, 2004) as a more appropriate measure for cluster separability.

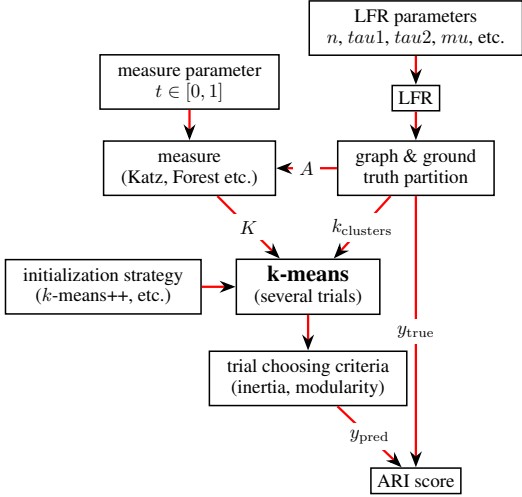

Figure 2: Measuring ARI clustering score for a particular graph, measure, and measure parameter

Thus, the defined set of features $[n, \tau_1, \tau_2, \text{avg degree}, \text{modularity}]$ is enough to consider all graph properties mentioned above. Although modularity is a widely used measure, it suffers from resolution limit problems (Fortunato & Barthelemy, 2007). We acknowledge that this may cause some limitations in our approach, which should be the topic of further research.

Table 2: The overall leaderboard. The win percentage is calculated among all 7396 graphs in the dataset. The ARI column shows the mean ARI across the dataset.

| # | Measure | Rank | Wins, % | ARI | # | Measure | Rank | Wins, % | ARI |
|---|---------|------|---------|-----|---|---------|------|---------|-----|
| 1 | SCCT | 4.7 | 56.0 | 0.58 | 14 | DF | 11.0 | 8.1 | 0.28 |
| 2 | NHeat | 6.9 | 25.2 | 0.40 | 15 | logAbs | 12.1 | 15.9 | 0.33 |
| 3 | RSP | 7.7 | 22.4 | 0.42 | 16 | logComm | 12.6 | 15.5 | 0.23 |
| 4 | SCT | 8.4 | 19.5 | 0.40 | 17 | logFor | 13.0 | 12.7 | 0.22 |
| 5 | Comm | 8.4 | 15.8 | 0.36 | 18 | HeatPR | 13.4 | 12.5 | 0.22 |
| 6 | logNHeat | 8.8 | 18.0 | 0.37 | 19 | logHeat | 13.6 | 11.9 | 0.21 |
| 7 | SP-CT | 9.0 | 20.4 | 0.41 | 20 | Heat | 14.7 | 11.2 | 0.19 |
| 8 | logHeatPR | 9.2 | 18.0 | 0.37 | 21 | logDF | 14.8 | 11.1 | 0.18 |
| 9 | FE | 9.5 | 19.3 | 0.39 | 22 | PPR | 16.7 | 4.0 | 0.13 |
| 10 | Katz | 9.8 | 7.7 | 0.32 | 23 | Abs | 18.6 | 3.7 | 0.08 |
| 11 | logKatz | 9.9 | 18.1 | 0.35 | 24 | For | 19.3 | 2.8 | 0.07 |
| 12 | logPPR | 10.0 | 17.5 | 0.35 | 25 | ModifPPR | 20.5 | 1.7 | 0.05 |
| 13 | logModifPPR | 10.5 | 17.3 | 0.35 | | | | | |

For every generated graph, we calculate the top ARI score for every measure (Hubert & Arabie, 1985). We choose ARI as a clustering score which is both popular and unbiased (Gösgens et al., 2019). As soon as every measure has a parameter, we perform clustering for a range of parameter values (we transform the parameter to become in the [0, 1] interval and then choose 16 values linearly spaced from 0 to 1). For each value, we run $6 + 6 + 6$ trials of $k$-means (6 trials for each of three initialization methods).

Fig. 2 shows the pipeline we use to calculate ARI score for a given LFR parameter set, a measure, and a measure parameter. Measure parameters are not the subject of our experiments, so for every measure we just take the result of the measure with the value of the parameter that gives the best ARI score.

Because of the need to iterate over graphs, measures, parameter values, and initializations, the task is computationally difficult. The total computation time was 20 days on 18 CPU cores and 6 GPUs.

## 4 RESULTS

### 4.1 GLOBAL LEADERSHIP IN LFR SPACE

We rank the measures by their ARI score on every graph of the dataset. The rank is defined as the position of the measure in this list, averaged over the dataset (see Table 2). It is important to note that the global leadership does not give a comprehensive advice on which measure is better to use, because for a particular graph, the global leader can perform worse than the others. Here we consider the entire LFR space, not just its zone corresponding to common real-world graphs, so the ranking may differ from those obtained for restricted settings.

As SCCT is the winner for both ranking and percentage of wins, we can say for sure that it is the global winner for the LFR space graphs. Other measures still can be leaders in some zones of the feature space.

### 4.2 FEATURE IMPORTANCE STUDY

First of all, we find out which graph features are important for the choice of the best measure and which are not. To do that, we use Linear Discriminant Analysis (Mika et al., 1999) (LDA). This method finds a new basis in the feature space to classify a dataset in the best way. It also shows how many components of basis are required to fit the majority of data.

Fig. 3a shows that the first two components take about 90% of the explained variance. Fig. 3b shows that these components include only $\tau_1$, avg degree, and modularity. The fact that $n$ is not used means that the size of the graph as well as the density are not of primary importance for choosing the best measure. So is not $\tau_2$ measuring the diversity of cluster sizes.

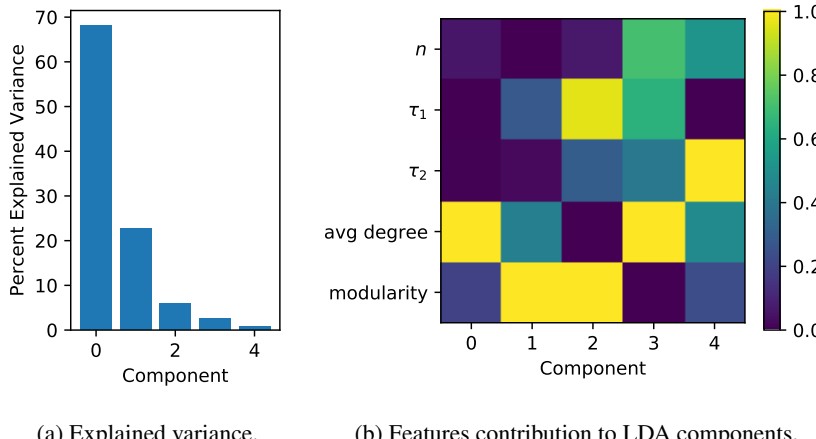

(a) Explained variance.          (b) Features contribution to LDA components.

Figure 3: The results of LDA components.

Fig. 4 shows the point cloud projected on the space of the two main components of LDA. We see a confirmation that the measures are indeed zoned, but the areas are quite noisy. To detect the zones of measure leadership, we need to know the leadership on average in every area of space, rather than the wins in particular points. To define the local measure leadership in the whole space, we need to introduce a filtering algorithm that for every point of the space returns the measure leadership depending on the closest data points. As the choice of measure is actually dependent only on three features, we can limit our feature space to $[\tau_1,$ avg degree, modularity$]$.

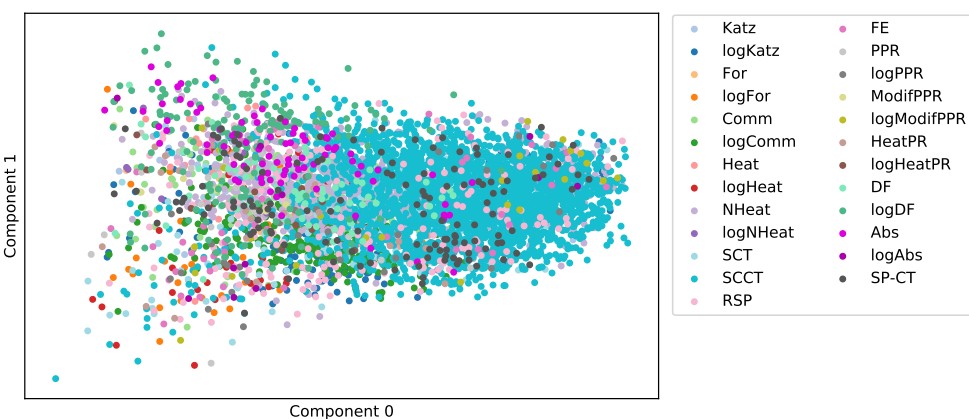

Figure 4: The dataset projected on the two main components of LDA; the point color represents the winning measure for each graph

### 4.3 GAUSSIAN FILTER IN FEATURE SPACE

Using a filter in the feature space, we can suppress the noise and find actual zones of leadership for the measures. We use the Gaussian filter with a scale parameter $\sigma$. For every given point of the space, it takes the data points that are closer than $3\sigma$ and averages ARIs of the chosen points with a weight $e^{-\text{dist}^2/2\sigma^2}$. This allows to give larger weights to closer points. If there are less than three data points inside the sphere with a $3\sigma$ radius, the filter returns nothing, allowing to ignore the points with insufficient data.

Before applying the filter, we prepare the dataset. First, we only take the points with only one winning measure, because multiple winners can confuse the filter. Then we normalize the standard

deviation of every feature distribution to one. Finally, we cut off the long tail of distant data points. The resulting number of graphs is 5201.

To choose $\sigma$, we apply the filter with different $\sigma$ and look at the number of connected components in the feature space. The needed $\sigma$ should be large enough to suppress the noise, however, it should not suppress small zones. Guided by this heuristic, we choose $\sigma = 0.5$.

Table 3: The leaderboard of measure wins after filtering with $\sigma = 0.5$

| Measure | SCCT | logComm | NHeat | Comm | logDF | RSP | FE | Abs | SCT | logNHeat | logFor |
|---|---|---|---|---|---|---|---|---|---|---|---|
| Wins | 4283 | 441 | 268 | 78 | 64 | 56 | 3 | 3 | 2 | 1 | 1 |

After filtering with $\sigma = 0.5$, the leaderboard of measure wins is changed (see Table 3). Only six measures keep their positions: SCCT, NHeat, logComm, Comm, logDF, and RSP. This means that these measures do have zones of leadership, otherwise they would be filtered out. We can plot the entire feature space colored by the leadership zones of the measures (Fig. 5). As the resulting space is 3D, we show slices of it by each of the three coordinates.

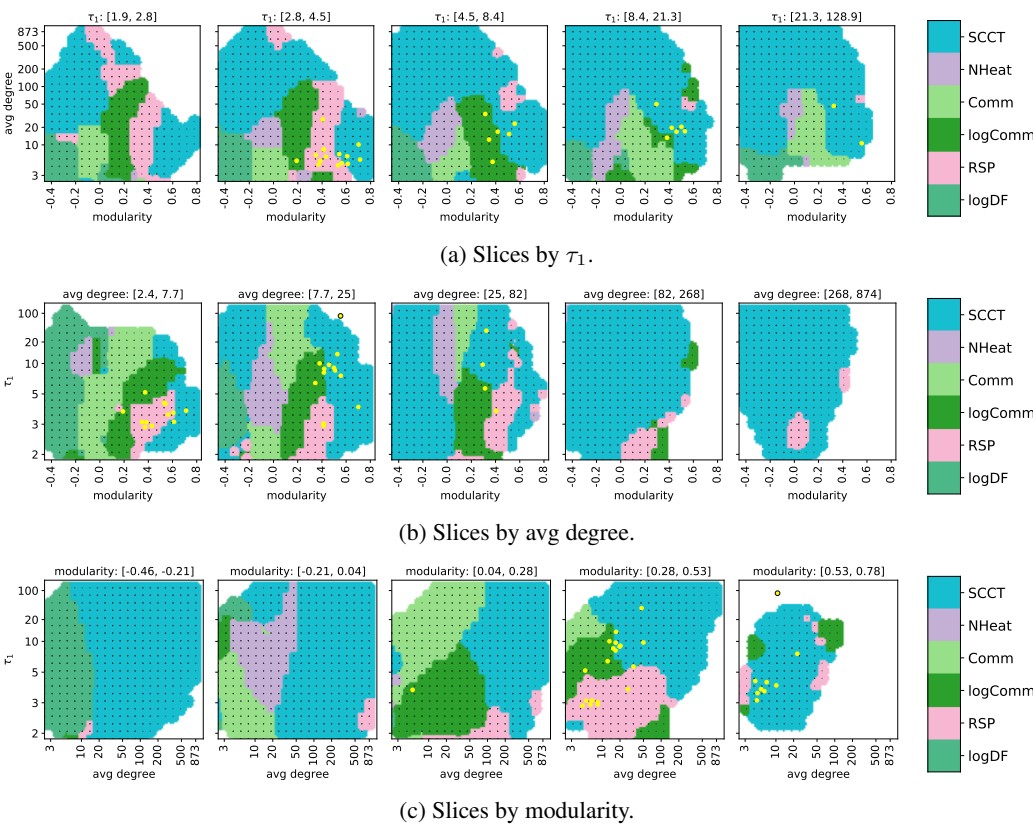

Figure 5: The feature space [$\tau_1$, avg degree, modularity] divided into the leadership zones of six measures. The yellow points represent the positions of real-world datasets under study in the space.

## 5 REAL-WORLD DATASETS

Even though LFR lacks some characteristics of real-world graphs, there is evidence that the optimal parameter of the Louvain clustering for a real graph is close to the parameter for LFR graphs gen-

erated from the features of a real one (Prokhorenkova, 2019). So, there is a chance that the learned space might be helpful for choosing measures in the wild.

For evaluation, we use 29 graphs of standard datasets: Dolphins (Lusseau et al., 2003), Football (Newman & Girvan, 2004), Karate club (Zachary, 1977), Newsgroups (9 subsets, weights are binarized with threshold 0.1) (Yen et al., 2007), Political blogs (Adamic & Glance, 2005), Political books (Newman, 2006), SocioPatterns Primary school day (2 graphs) (Stehlé et al., 2011), Cora (11 subsets) (McCallum et al., 2000), Eu-core (Leskovec et al., 2007), EuroSIS (WebAtlas, 2009). The parameters of these graphs are marked in Fig. 5. For each graph, we found the best ARI for every measure (iterating over the measure parameter value). Now we can check the quality of measure choice, based on the found LFR data. The result of LFR recommendation is the measure that is chosen for the set of parameters corresponding to the dataset in hand.

Table 4: Mean ARI of the LFR recommended strategies for datasets. Top6 stands for the set of measures that have their zones in the LFR parameter space.

| Strategy | Mean ARI |
|---|---|
| Always take SCCT | 0.61 |
| Based on LFR space, top6 measures | 0.62 |
| Based on LFR space, all measures | 0.62 |
| Upper bound | 0.64 |

The best measures on the considered datasets are SCCT (by the mean ARI) and SCT (by the rank). This is pretty similar to the results obtained for LFR. Moreover, Spearman correlation between the ranks of measures for the datasets and for the corresponding LFR recommendations is 0.90.

Let us use "always take SCCT" as our baseline strategy. In Table 4 we compare it with strategies based on the LFR space. We obtain LFR recommendation using $k$nn as a well-proven method for meta-learning. Since each graph is unique, the result of 1nn can be very noisy, thus we use 5nn.

Table 4 shows that the recommendation approach slightly beats the baseline. Reducing the number of measures from 25 to 6 do not drop the quality. However, this quality increase is not enough to draw confident conclusions about the advantages of the method. Using this fact and the fact that the ranks on datasets and recommendations are highly correlated, we conclude that the meta-learning procedure is adequate to give a robust recommendation, but not precise enough to beat the baseline confidently. This may be due to the fact that that the nodes of real graphs were not labeled systematically since they were created in the wild. A larger dataset could help separate the signal from the noise and pinpoint where the limits of the method are. At least, the good news is that the conclusions made on the LFR basis do not contradict the results obtained on the datasets.

## 6 Conclusions

In this work, we have shown that the global leadership of measures does not provide comprehensive knowledge about graph measures. We demonstrated that among 25 measures, SCCT is the best measure for the LFR graphs both by winning rate and ranking. However, there are also smaller confident zones of leadership for NHeat, Comm, logComm, logDF, and RSP.

Our results do not contradict those of other experimental works and rather expand them by providing new findings. LogComm was first introduced in Ivashkin & Chebotarev (2016) and won in the competitions on graphs generated with a fixed set of SBM parameters. This study confirms its leadership, but only for a certain type of graphs. Another interesting finding is logDF, which unexpectedly shows good performance for the graphs with low modularity and low average degree.

This study is based on the LFR benchmark data. An attempt to apply the results to real data gives the quality slightly above the baseline. However, there is a strong correlation between the ranking of measures for datasets and the ranking of LFR recommendation, which indicates that the leading measures are the same, while the recommendations are not precise enough.

It can be noted that our study is insensitive to the non-uniformity of the generated dataset. While manipulations with this dataset may affect the global leaderboard, they cannot change the local leadership studied in this work.

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
