# OpenReview forum: "Dissecting graph measures performance for node clustering in LFR parameter space"
_ICLR.cc/2021/Conference — Reject_

### Official Review · AnonReviewer1 · 2020-10-19
**An interesting yet limited comparative review**

**Rating:** 6
**Confidence:** 4

**Review:**

Using 7500 LFR-generated graphs as a benchmarks suite, the authors compare 25 graph clustering measures, determining the best measure for every area of the parameter space. The paper is well written, mathematically sound and interesting, and definitely useful to the graph theory community. However, as acknowledged by the authors, the study is limited by the structure of the benchmark suites, which is restricted to networks that can be generated by LFR rules. Overall, I rate it as a weak accept.

Pros:
- the analysis is clear and grounded, with a sufficient level of mathematical details
- the authors point out a clear winner out of the set of compared metrics

Cons:
- the amount of novelty in the manuscript is limited
- the benchmark suite is very specific, and no real world example (that would have added great value to the submission) is provided

---

> ### Author Response · Authors · 2020-11-18
> **Response to Reviewer #4**
>
> We would like to thank AnonReviewer1 for providing constructive suggestions, which helped to improve the manuscript. We have addressed all the comments below.
>
> _REVIEWER 1: Using 7500 LFR-generated graphs as a benchmarks suite, the authors compare 25 graph clustering measures, determining the best measure for every area of the parameter space. The paper is well written, mathematically sound and interesting, and definitely useful to the graph theory community._
>
> AUTHORS: Thanks!
>
> _COMMENT 1: However, as acknowledged by the authors, the study is limited by the structure of the benchmark suites, which is restricted to networks that can be generated by LFR rules._
>
> RESPONSE: A possible way of assessing the submitted paper is comparing it with the previous works in the same area, e.g., [1–8] (see the references below). Despite the fact that the idea of measure comparison on k-means was already tested in Yen et al., 2009 [1], there are still no complete results on a noticeable number of measures. We propose a new step in understanding the measures’ applicability to clustering problems, namely, we:
> 1. Systematically study, for the first time, not only overall (average) leadership of measures, but dependence of their performance on the graph parameters: the power law degree distribution, average degree, and modularity;
> 2. Consider a much wider class of vertex closeness measures consisting of 25 parametric families of measures, basically, all the most popular families we know;
> 3. Combine several approaches to k-means seeding;
> 4. Check which graph parameters do and which do not affect the choice of optimal measure.
> The corresponding experiment was rather massive. As a result of applying k-means with 18 initializations for every measure family (of 25) with 16 parameter values on 7396 graphs, the pure computation time was 20 days on 18 CPU cores and 6 GPUs.
> Of course, the universe of LFR graphs is restricted. However, it is a nontrivial 5-parametric (with extra parameters) class, which is much more diverse, than, say, the SBM class. While it is not easy at all to approach considering an unrestricted class of networks, in the previous literature, the sets of benchmark suites were narrower.
>
> _COMMENT 2: Pros:_
> * _the analysis is clear and grounded, with a sufficient level of mathematical details_
> * _the authors point out a clear winner out of the set of compared metrics_
> _Cons:_
> * _the amount of novelty in the manuscript is limited_
>
> RESPONSE: The authors believe that the novelty of the research lies in the above 1-4.
>
> _COMMENT 3:_
> * _the benchmark suite is very specific, and no real world example (that would have added great value to the submission) is provided_
>
> RESPONSE: To check whether the results are reproducible on real graphs, we added a number of real-world examples to the next revision of the paper!
>
> References:
> 1. The following papers are from the reference list of our manuscript: Fouss et al. (2006), Yen et al. (2007, 2008, 2009), Kivimäki et al. (2014), Ivashkin et al. (2016), Sommer et al. (2016, 2017), Avrachenkov et al. (2017).
> 2. F. Fouss et al. (2012) Neural Networks. V. 31. P. 53-72.
> 3. G. Guex, et al. (2018) arXiv:1806.03232
> 4. G. Guex, et al. (2019) arXiv:1902.03002
> 5. R. Aynulin (2019) Int. Conf. Complex Networks and Their Applications. Springer, P. 188-197.
> 6. R. Aynulin (2019) Int. Workshop Algorithms and Models for Web-Graph. Springer, P. 16-29.
> 7. S. Courtain, et al. (2020) Information Sciences. https://doi.org/10.1016/j.ins.2020.10.005
> 8. P. Leleux, et al. (2020) arXiv:2007.00419

---

### Official Review · AnonReviewer4 · 2020-10-26
**The paper empirically evaluate the distance measure between nodes in a graph under the setting of kernel k-means and LFR data generation.**

**Rating:** 5
**Confidence:** 2

**Review:**


1. No good reasons to choose settings of evaluation, particularly kernel k-means and LFR.

2. I think this paper may think about something very obvious. The setting is kernel k-means, and if the similarity measure is given as some kernel, it might be more clearly shown what kernel is good under what condition, under kernel k-means.  In other words, we may find some connection between a kernel and the condition of data generation under the kernel k-means already before doing some experiments.  I think this type of investigation is missing in this paper.

3. So what is the reason why SCCT is the best and/or why highly-ranked methods are so? I think that would be simply connected to the scheme of data generation of LFR or another feature in generating data or noise. I think this point might be obvious but might become some good contribution, while just data generation and comparison would not be something people can say contribution.

---

> ### Author Response · Authors · 2020-11-18
> **Response to Reviewer #4 (part 1)**
>
> We would like to thank Referee #4 for providing valuable comments and suggestions, which helped improve the revised manuscript. We have addressed all the comments below.
>
> _COMMENT 1: No good reasons to choose settings of evaluation, particularly kernel k-means and LFR_
>
> RESPONSE: Kernel k-means is used here because it is ideal for our purpose: to compare the usefulness of various node closeness measures in graph clustering problems. LFR was chosen as it produces a fairly diverse family of graphs with clustering structure. Kernel k-means + LFR allow us to test a novel methodology for finding areas of measure leadership in a parameter space and positioning real graphs on this “map.” This methodology can then be applied to alternative settings in order to combine particular results into more general ones.
> We have expanded the explanation of motivation in the revised version of the paper to clarify the work's place in the experimental graph measure studies.
>
> _COMMENT 2a: I think this paper may think about something very obvious. The setting is kernel k-means, and if the similarity measure is given as some kernel, it might be more clearly shown what kernel is good under what condition, under kernel k-means._
>
> RESPONSE: We do not know any reliable evidence of this kind for a natural and complex enough node closeness measure. In addition (or, more like, multiplication) to 25 measure families, we have a five-parametric LFR class of graphs. Besides, every measure family has a parameter which affects the properties of the measure. An indirect confirmation that the multipart problem of assessing each measure over the entire parameter space is not so obvious is the fact that there is an extensive literature where such problems are studied experimentally, for example, [1-8]. Our paper is a continuation of this sequence. While global ranking suffers from data distribution issues, our zone leadership study is resistant to it. Moreover, it gives answers to questions, which were not covered by previous work:
> ●	How strongly does the choice of measure depend on the properties of the graph?
> ●	Which graph properties are the most important?
> ●	Which non-leading measures need further consideration, and which do not?
> Nonetheless, as our work lacked persuasiveness without real-world data, we added a chapter about applying the results received on LFR to the real-world graphs in the next revision.
>
> _COMMENT 2b: …we may find some connection between a kernel and the condition of data generation under the kernel k-means already before doing some experiments. I think this type of investigation is missing in this paper._
>
> RESPONSE: This would be a very natural way of investigation; however, it is not easy at all. Such results require a quite subtle reasoning. As already said, many of the 25 families of closeness measures we compare are quite complicated, and their properties depend on their own parameters. One of the few results that resemble the desired conclusions says that the commute time distance performs poorly for large graphs [9]. Such proofs are rather difficult; however, it is an interesting road for future research. Thus, our present work extends the empirical base of theoretical results: it helps to find out what exactly requires proof. The purpose of this paper is to bring together all of the commonly used node closeness measures and go beyond global rankings (which are highly data dependent) to the areas of measure leadership. We added the corresponding remark to the paper.
>
> _COMMENT 3: So what is the reason why SCCT is the best and/or why highly-ranked methods are so? I think that would be simply connected to the scheme of data generation of LFR or another feature in generating data or noise. I think this point might be obvious but might become some good contribution, while just data generation and comparison would not be something people can say contribution._
>
> RESPONSE: We agree: this point will be a very good contribution. However, this is not easy; as a result, experimental investigation of such problems has a rather long history. The authors believe that they are halfway there. At the present stage, they radically narrow the class of promising measures and determine what exactly requires proof. At the next stage, they will prove the found hypotheses. We added the corresponding note to the revised version of the paper.

---

> > ### Author Response · Authors · 2020-11-18
> > **Response to Reviewer #4 (part 2)**
> >
> > References:
> > 1. The following papers are from the reference list of our manuscript: Fouss et al. (2006), Yen et al. (2007, 2008, 2009), Kivimäki et al. (2014), Ivashkin et al. (2016), Sommer et al. (2016, 2017), Avrachenkov et al. (2017)
> > 2. F. Fouss et al. (2012) Neural Networks. V 31. P 53-72
> > 3. G. Guex, et al. (2018) arXiv:1806.03232
> > 4. G. Guex, et al. (2019) arXiv:1902.03002
> > 5. R. Aynulin (2019) Int. Conf. Complex Networks and Their Applications. Springer, P 188-197
> > 6. R. Aynulin (2019) Int. Workshop Algorithms and Models for Web-Graph. Springer, P 16-29
> > 7. S. Courtain, et al. (2020) Information Sciences. https://doi.org/10.1016/j.ins.2020.10.005
> > 8. P. Leleux, et al. (2020) arXiv:2007.00419
> > 9. U. von Luxburg, A. Radl, M. Hein (2010). Proc. of NIPS’10 Conf. P 2622–2630

---

### Official Review · AnonReviewer3 · 2020-10-28
**Interesting empirical analysis of different graph measures for graph clustering. The choice of LFR though limits the generalization of the observations. Also, the proposed approach lacks theoretical justification.**

**Rating:** 3
**Confidence:** 5

**Review:**

The paper deals with the problem of community detection on graphs, examining the impact of graph measures. To do so, the paper proposes an experimental framework where clustering is achieved using the kernel k-means algorithm, and the performance of graph measures is examined on various instances of artificially generated graphs using the LFR benchmark. The overall approach is empirical, supported mainly by the experimental results. The main observations concern the consistent behavior of particular graph measures across multiple settings of the dataset.

Strong points:

-- The paper addresses an important problem in network analysis, which also concerns practitioners in a wide range of disciplines.

-- Various graph measures are considered in the evaluation. This is very interesting, since, in my view, some of them are not very well-known among the graph clustering / community detection communities.

-- The paper is well-structured and well-written. Most of the concepts, including the experimental framework, are clearly presented.


Weak points:

--- My main concern about the paper has to do with the consistency of the proposed evaluation framework under different evaluation criteria and graph data beyond the LFR benchmark. Firstly, as the paper also mentions, focusing only on LFR graphs can definitely reveal important properties of algorithms, but limits the generalization of the observations in the case where other generators might be used (e.g., SBM) or even real-world graphs. Besides, the argument made in the paper that the LFR benchmark generates graphs similar to real-world ones is not very accurate. LFR focuses on the clustering structure as well as on the degree distribution but might miss other key properties including graph diameter or the number of triangles (or the clustering coefficient). Is there any evidence that could support the observations of the paper in the case of real graphs?

--- A closely related point has to do with the observation that real-world graphs do not have a clear clustering structure, i.e., well-defined cuts. Focusing only LFR graphs might not be enough to capture such instances.

--- The modularity criterion is used to evaluate the quality of communities, which overall is a widely used criterion. Nevertheless, modularity has been shown to be prawn to the particular structure of the communities (e.g., resolution limit). How is this taken into account in the evaluation of the communities?

--- Another point is that the paper is purely empirical. Definitely, this is a good starting point, especially when the focus is on experimental settings that have not been used before. Nevertheless, despite the interesting observations, I would expect to have a (theoretical) justification or some reasoning of why SCCT performs well on LFR graphs, or for instance why PPR which is a widely used measure, shows poor behavior.

--- In the paper, all graph measures are considered as kernels. How valid is this argument? Why not just using the pure k-means for graph measures which are not kernels?

---

> ### Author Response · Authors · 2020-11-18
> **Response to Reviewer #3 (part 1)**
>
> We would like to thank Referee #3 for providing valuable comments and suggestions, which helped improve the manuscript. We have addressed all of them below.
>
> _REVIEWER 3:  Strong points:
> -- The paper addresses an important problem in network analysis, which also concerns practitioners in a wide range of disciplines.
> -- Various graph measures are considered in the evaluation. This is very interesting, since, in my view, some of them are not very well-known among the graph clustering / community detection communities.
> -- The paper is well-structured and well-written. Most of the concepts, including the experimental framework, are clearly presented._
>
> AUTHORS: Thanks!
>
> _COMMENT 1: Weak points:
> --- My main concern about the paper has to do with the consistency of the proposed evaluation framework under different evaluation criteria and graph data beyond the LFR benchmark. Firstly, as the paper also mentions, focusing only on LFR graphs can definitely reveal important properties of algorithms, but limits the generalization of the observations in the case where other generators might be used (e.g., SBM) or      even real-world graphs._
>
> RESPONSE: We agree. On the other hand, a possible way of assessing the submitted paper is comparing it with the previous work in the same area, e.g., [1–8] (the references are given below). It can be observed that the sets of benchmark suites of those papers are more restricted. These are either SBM graphs (Ivashkin et al., 2016), or individual LFR graphs, or just a few examples of real-world graphs (since the available set of test networks with marked non-overlapping communities is rather scarce). It is remarkable that our results on the performance of node closeness measures are consistent with the previous ones (the corresponding comparison has been added to the paper). Thus, we do not pretend to present the universal final result, which is a matter for the future; we rather essentially expand the experimental scope of the previous work by including the rather diverse and non-trivial LFR family.
> In addition, we systematically study, for the first time, not only overall (average) leadership of measures, but dependence of their performance on the specific properties of graphs. In the revised version, we have implemented a related approach, in which for a given real-world network, we find its “representatives,” i.e., the closest graphs from a well-studied family and then apply the clustering method optimal for its representatives to the network itself.
>
> _COMMENT 2: Besides, the argument made in the paper that the LFR benchmark generates graphs similar to real-world ones is not very accurate. LFR focuses on the clustering structure as well as on the degree distribution but might miss other key properties including graph diameter or the number of triangles (or the clustering coefficient). Is there any evidence that could support the observations of the paper in the case of real graphs?_
>
> RESPONSE: You are right. We have clarified our statement. The LFR class is more diverse and realistic than the other cluster-based family, SBM. There is also work showing that the LFR-approximation is useful for determining algorithm parameters for real graphs [10]. Thus, even though LFR misses some important features of real graphs, it can still be useful in meta-learning.
> We have now included the investigation of a number of real graphs in the revised version of the paper. And the relationship between the results is quite interesting.
>
> _COMMENT 3: --- A closely related point has to do with the observation that real-world graphs do not have a clear clustering structure, i.e., well-defined cuts. Focusing only LFR graphs might not be enough to capture such instances._
>
> RESPONSE: This is an interesting but rather subtle problem, since the clustering structure of LFR graphs with some combinations of parameters is also quite difficult to identify.
>
> _COMMENT 4: --- The modularity criterion is used to evaluate the quality of communities, which overall is a widely used criterion. Nevertheless, modularity has been shown to be prawn to the particular structure of the communities (e.g., resolution limit). How is this taken into account in the evaluation of the communities?_
>
> RESPONSE: This also looks like an interesting but rather subtle problem. The logic of our study is as follows: obtain a result with widely used criteria at the first stage and test its robustness at the second one. And now we are on the first stage.

---

> > ### Author Response · Authors · 2020-11-18
> > **Response to Reviewer #3 (part 2)**
> >
> > _COMMENT 5: --- Another point is that the paper is purely empirical. Definitely, this is a good starting point, especially when the focus is on experimental settings that have not been used before. Nevertheless, despite the interesting observations, I would expect to have a (theoretical) justification or some reasoning of why SCCT performs well on LFR graphs, or for instance why PPR which is a widely used measure, shows poor behavior._
> >
> > RESPONSE: This is definitely the starting point. Although the field is covered by a number of works (some references have been mentioned above), theoretical results have not yet been achieved here, as they require a quite subtle reasoning. One of the few results of similar nature says that the commute time distance performs poorly for large graphs [9]. Such proofs are rather difficult; however, it is an interesting road for future research. Thus, our present work extends the empirical base of theoretical results: it helps to find out what exactly requires proof. The purpose of this paper is to bring together all of the commonly used node closeness measures and go beyond global rankings (which are highly data dependent) to the areas of measure leadership. We added the corresponding remark to the paper.
> >
> > _COMMENT 6: --- In the paper, all graph measures are considered as kernels. How valid is this argument? Why not just using the pure k-means for graph measures which are not kernels?_
> >
> > RESPONSE: It was shown that using, within the same framework, matrices that are not positive semidefinite (= are not valid kernels) can be effective (Yen et al., 2009). These matrices can also be used in pure k-means, but in this case we ignore their closeness nature. For these reasons, we decided to use the current setup as more intuitive for measure comparison. We added a related note.
> >
> > References:
> > 1. The following papers are from the reference list of our manuscript: Fouss et al. (2006), Yen et al. (2007, 2008, 2009), Kivimäki et al. (2014), Ivashkin et al. (2016), Sommer et al. (2016, 2017), Avrachenkov et al. (2017)
> > 2. F. Fouss et al. (2012) Neural Networks. V 31. P 53-72
> > 3. G. Guex, et al. (2018) arXiv:1806.03232
> > 4. G. Guex, et al. (2019) arXiv:1902.03002
> > 5. R. Aynulin (2019) Int. Conf. Complex Networks and Their Applications. Springer, 188-197
> > 6. R. Aynulin (2019) Int. Workshop Algorithms and Models for Web-Graph. Springer, 16-29
> > 7. S. Courtain, et al. (2020) Information Sciences. https://doi.org/10.1016/j.ins.2020.10.005
> > 8. P. Leleux, et al. (2020) arXiv:2007.00419
> > 9. U. von Luxburg, A. Radl, M. Hein (2010). Proc. of NIPS’10 Conf. P 2622–2630
> > 10. L. Prokhorenkova (2019). Using synthetic networks for parameter tuning in community detection. In International Workshop on Algorithms and Models for the Web-Graph. Springer. 1-15

---

### Official Review · AnonReviewer2 · 2020-10-30
**Unrealistic setting**

**Rating:** 4
**Confidence:** 5

**Review:**

This paper tests performance of different node similarity measures when used in K-means for clustering LFR graphs. It provides recommendations for which measure is more appropriate for different parameter spaces.

It is easy to read paper and well organized. A rich set of graph similarity measures is studied.
There are however critical issues with the design of the experiments. First, the space of parameters considered is very unrealistic and hence most experiments and recommendations are not applicable. This makes, for example, most of the space in figure 6, irrelevant. For instance the exponent of the degree distribution is between 2 and 3 for most real world networks. This goes up to 100 in the experiments. Where did you use the transformed [0,1] version?
The average degree is much smaller whereas graphs are usually much larger than the setting here. A negative modularity is result of poor parameter choices, and generally, signifies there is no cluster structure in the graph. Generally, given this is focused on studying the space of LFR graphs, one expects more careful understanding of what there parameters mean. On a minor point, LFR is not using preferential attachment mechanism and is achieving power-law degree distribution by directly sampling degrees and using configuration model.
I suggest fixing the parameter settings and making sure the modularity is at least 0.1 for all the graphs. The common link prediction measures, e.g. number of common neighbours, could also be added as simple baselines, as well as more recent embedding based models where k-means could be applied directly in the embedded space. Another possibility is to also include another clustering measure, say a density based one. The motivation needs to be expanded. What justifies going n square when a graph based clustering could do the same job? Also in the results, using ranking is good, but the actual numbers are also meaningful. It is hard to see if these methods are recovering anything meaningful when only the relative performance is reported. Basically, if the ARI is too small, which means no correlation between results and the ground-truth, you can still have a ranking of all the close to zero numbers.

---

> ### Author Response · Authors · 2020-11-18
> **Response to Reviewer #2**
>
> We would like to thank AnonReviewer2 for providing many valuable comments and suggestions, which helped improve the clarity of the revised manuscript. We have addressed all of them below.
>
> _COMMENT 1: It is easy to read paper and well organized. A rich set of graph similarity measures is studied. There are however critical issues with the design of the experiments. First, the space of parameters considered is very unrealistic and hence most experiments and recommendations are not applicable. This makes, for example, most of the space in figure 6, irrelevant. For instance the exponent of the degree distribution is between 2 and 3 for most real world networks. This goes up to 100 in the experiments. …The average degree is much smaller._
>
> RESPONSE: The authors believe that if they pay proper attention to practical settings, then they may pay some attention to theoretical settings too. Under the transformation we use, the exponent of the degree in [2, 3] gets about 15% of space and the same fraction of samples in the dataset, which is enough for reliable results. Since the realistic range is covered, its expansion is not a big drawback.
> The Referee is right: our interest lies not just in revealing the best measures for well-known applications, but also in uncovering the best measures for diverse types of graphs, some of which may appear tomorrow in specific applications. A secondary question is: if a measure never proved to be useful, can we exclude it from future research? This will allow us to study the perspective measures in more detail.
> We added a remark on realistic settings.
>
> _COMMENT 2: Where did you use the transformed [0,1] version?_
>
> RESPONSE: For all the experiments (and for parameter sampling of dataset), the parametric range of power law exponents (tau1, tau2) is transformed with 1 - 1/sqrt(x), which takes [1, infty] onto [0, 1]. For Gaussian filtering (section 4.3), we additionally normalize all features to equalize their influence on the smoothing result.
>
> _COMMENT 3: The… graphs are usually much larger than the setting here._
>
> RESPONSE: Generating much larger graphs would make the experiment unfeasible for the time being. The pure computation time for this work was 20 days on 18 CPU cores and 6 GPUs. Fortunately, according to our results, the number of nodes is not among the most significant features for the choice of measure, at least for LFR with the number of nodes up to 1500. That is why we decided not to increase the number of nodes. We added a remark on the size to the submission.
>
> _COMMENT 4: A negative modularity is result of poor parameter choices, and generally, signifies there is      no cluster structure in the graph._
>
> RESPONSE: The case of negative modularity may occur when the frequency of inter-class edges is higher than that of intra-class edges. Such bipartite/multipartite-like structures appear in many applications, and revealing them is an interesting problem. We have found in our experiments that the logDF is a stable leader in that area: it performs clearly better than the others. It looks like an impressive result, so we decided to present it in the paper. We added a note regarding this.
>
> _COMMENT 5: Generally, given this is focused on studying the space of LFR graphs, one expects more careful understanding of what there parameters mean._
>
> RESPONSE: Yes, in the revised version, we explain the parameters in more detail, thanks.
>
> _COMMENT 6: On a minor point, LFR is not using preferential attachment mechanism and is achieving power-law degree distribution by directly sampling degrees and using configuration model._
>
> RESPONSE: Thanks, we now mention this.
>
> _COMMENT 7: I suggest fixing the parameter settings and making sure the modularity is at least 0.1 for all the graphs._
>
> RESPONSE: We still kept an interval of negative modularity in order not to rule out the presumptive case of bipartite/multipartite-like networks. This case may require specific measures, however finding universal measures most tolerant to this case is a worthy task.
>
> _COMMENT 8: The common link prediction measures, e.g. number of common neighbours, could also be added as simple baselines, as well as more recent embedding based models where k-means could be applied directly in the embedded space. Another possibility is to also include another clustering measure, say a density based one._
>
> RESPONSE: Yes, the work can be extended by adding extra graph features or clustering measures, or methods. That's the right thing to do, but we already compare 25 families measures in the current work. Our goal here is not to find the best clustering approach among all the methods, but rather to propose and test—within the framework of some popular approach to clustering—a methodology for finding areas of measure leadership and positioning real graphs on this “map.” This methodology can then be applied to alternative settings in order to combine particular results into more general ones.

---

> > ### Author Response · Authors · 2020-11-18
> > **Response to Reviewer #2 (part 2)**
> >
> > _COMMENT 9: The motivation needs to be expanded. What justifies going n square when a graph based clustering could do the same job?_
> >
> > RESPONSE: This work may be considered in the context of experimental studies of the performance (mainly in clustering tasks) of node closeness measures. Early papers of this direction are Fouss et al. (2006), Yen et al. (2007, 2008, 2009), then Kivimäki et al. (2014), Ivashkin et al. (2016), Sommer et al. (2016, 2017), Avrachenkov, et al. (2017)—all these papers are from the reference list of our manuscript—and many others. Basically, a number of these papers have similar motivations. The major difference of the present work is that we consider a much wider set of 25 parametric families of measures and search, for the first time, the parametric space areas of measure leadership instead of just their average performance in some settings. We have added a clarifying note.
> >
> > _COMMENT 10: Also in the results, using ranking is good, but the actual numbers are also meaningful. It is hard to see if these methods are recovering anything meaningful when only the relative performance is reported. Basically, if the ARI is too small, which means no correlation between results and the ground-truth, you can still have a ranking of all the close to zero numbers._
> >
> > RESPONSE: Thanks, we added some actual numbers, which add to the meaning of stable rankings of measures.

---

### Decision · Program_Chairs · 2021-01-11
**Final Decision**

**Decision:**

Reject

**Comment:**

This paper studies various graph measures in depth.  The paper was reviewed by three expert reviewers who complemented the ease of understanding because of clear writing. But they also expressed concerns for limited novelty, theoretical justification, and unrealistic setting. The authors are encouraged to continue research, taking into consideration the detailed comments provided by the reviewers.